# Maturation of Arousals during Day and Night in Preterm Infants

**DOI:** 10.3390/children9020223

**Published:** 2022-02-08

**Authors:** Aurore Guyon, Francoise Ravet, Alex Champavert, Marine Thieux, Hugues Patural, Sabine Plancoulaine, Patricia Franco

**Affiliations:** 1Unité de Sommeil, Service D’épilepsie, Sommeil et Explorations Fonctionnelles Neurologiques Pédiatriques—INSERM U1028—Hôpital Femme-Mère-Enfant, Hospices Civils de Lyon, Université Claude Bernard Lyon 1, 59 bd Pinel Bron, 69500 Lyon, France; aurore.guyon@chu-lyon.fr (A.G.); ext-alex.champavert@chu-lyon.fr (A.C.); marine.thieux@chu-lyon.fr (M.T.); 2Unité de Sommeil Pédiatrique—Département Universitaire de Pédiatrie du CHU Liège—Site CHR, 1 bd du XIIème de Ligne, 4000 Liège, Belgium; sommeil.ravet@gmail.com; 3Neonatal and Pediatric Intensive Care Department, Inserm, U1059, University Hospital of Saint Etienne, 42270 Saint Etienne, France; hugues.patural@chu-st-etienne.fr; 4Centre of Research in Epidemiology and StatisticS (CRESS), Inserm, INRAE, Université de Paris Cité, F-75004 Paris, France; sabine.plancoulaine@inserm.fr

**Keywords:** sleep, arousals, preterm, SIDS

## Abstract

The objective of this study was to compare the maturation of spontaneous arousals during day and night sleep in preterm and term infants. From the Autonomic Baby Evaluation study, the sleep and arousal characteristics of 12 preterm (35.1 ± 2.1 weeks’ gestational age, GA) and 21 term (39.8 ± 0.8 weeks GA) newborns were compared between diurnal and nocturnal sleep periods at birth (M0) and 6 months (M6) of age. Models were adjusted for time (night/day), maturation (M0/M6), prematurity (yes/no). We found that preterm infants had less active sleep (AS)% than term infants with maturation during both day and night sleep, which may reflect accelerated brain maturation secondary to stress or environmental exposure after birth. Moreover, there was a difference in arousal maturation during day and night sleep in the preterm infants, as shown previously for term infants, which suggests the emergence of a circadian rhythm during the earliest postnatal period. We also showed that compared to term infants, these moderate preterm infants had fewer total arousals and, more specifically, fewer arousals in AS during day and night sleep, exposing them to a higher risk of sudden infant death syndrome.

## 1. Introduction

Arousal from sleep could play a central role in protection against the respiratory or cardiovascular challenges that may occur during sleep [1]. Several studies suggested that a failure to arouse could be involved in the final steps of SIDS [2,3,4]. From epidemiological studies, Filiano and Kinney proposed a triple risk model for SIDS combining three critical factors: prenatal vulnerability, an exogenous postnatal stressor, and a critical developmental period [5]. Prenatal vulnerability could be secondary to an adverse intrauterine environment (such as in-utero growth retardation, exposure to maternal tobacco smoking, or drug addiction) or to premature birth [6,7,8,9]. Worldwide, about 8 to 10% of births occur before 37 weeks of gestation [10], and these preterm infants represent 29 to 34% of SIDS victims [11,12,13].

Sleep is closely related to development and is established from the fetal period: an activity/rest rhythm can be detected at 20–22 weeks of gestation, and stages of active sleep (AS) and quiet sleep (SC) appear between the 27th and 30th weeks of gestation [14]. At birth, newborns fall asleep in AS. Their sleep cycles last 50 to 60 min and are composed of AS (representing 50 to 60% of the sleep time) followed by QS (30 to 40%) and, finally, an indeterminate or transitional sleep period (10 to 20% of the cycle period).

Over 24 h, sleep is divided into periods of 3 to 4 h, following an ultradian rhythm. According to some authors, the day–night distinction is not present at birth but appears following the maturation of the circadian rhythm [15]. Conversely, Rivkees et al. suggested that the biological clock appears during the fetal period [16]. Jenni et al. also showed that starting from the second or third week of life, most full-term infants develop day–night asymmetry in terms of sleep (i.e., longer sleep periods during the night than the day) [17]. A few other studies also showed that full-term infants on whom a particular meal rhythm was imposed, and who were exposed to sunlight and darkness at night, acquired a day–night rhythm more quickly [18]. Although one could expect the sleep–wake rhythms of preterm infants to be identical to gestational-age-matched in utero fetuses, the differences observed suggest these rhythms are influenced by extra-uterine life. In neonatal intensive care services, the exposure of preterm infants to light during the day (239 ± 29 lux from 7 a.m. to 7 p.m.) and to darkness at night induced cycles of rest–activity that were present for as long as 1 month after discharge from hospital. By contrast, preterm infants who were continuously exposed to low light (<25 lux) did not present any type of rest–activity cycle before 3–4 weeks after discharge. These studies show that cyclic exposure to light (the day–night effect) stimulates the circadian clock and therefore allows a circadian rhythm of activity to appear earlier [19].

The arousability of infants can be influenced by the duration of gestation. Indeed, preterm infants (born at 31–35 weeks of gestation) display a delay in the maturation of sleep-state-related difference in arousability between AS and QS compared to healthy-term infants (born at 37–42 weeks of gestation); this difference usually appears 2–3 weeks after birth, but only 2–3 months after birth for these preterm infants [20]. Indeed, during infancy, spontaneous arousal and awakening occurs more frequently in AS compared to QS [21,22]. Furthermore, in AS, the frequency of movements and arousals is greater compared to QS, although this difference is smaller in neonates than in one-month-old infants [23]. Moreover, decreased arousal responses have been reported in preterm infants (26–32 weeks gestation) with a history of apnea and bradycardia, both in QS and AS at term, and in QS only 2–3 months after birth [24]. Preterm infants display a longer arousal latency in AS at 2–3 weeks of age and reach significantly lower SpO_2_ levels at 2–3 weeks in both AS and QS and at 2–3 months in QS after a mild hypoxia challenge (15% oxygen; preterm infants were all evaluated at corrected gestational age) [25]. Considering this greater desaturation during hypoxic challenge, as well as the longer arousal latency, one can hypothesize that preterm infants have an impaired or inadequate response to hypoxia during sleep, which may be related to the greater risk of preterm infants suffering from SIDS. Interestingly, preterm infants die of SIDS at an earlier corrected age compared to term infants [26].

Arguing that no difference exists in the maturation of arousals during the nychthemere, different authors have conducted studies to investigate SIDS risk factors, regardless of the time of day. Recently, the maturation of spontaneous arousals was compared in term newborns during the day and during the night: during the day, there was a lower total sleep time (TST) and lower sleep efficiency (SE) with less QS and AS compared to during the night [27]. However, there were more arousals, especially in AS (AAS) during the day than during the night. With maturation, TST, SE, and AAS decreased during the day and increased during the night. The infants also had a more marked increase in QS and decrease in AS during the day than during the night.

The objective of this study was to compare the maturation of spontaneous arousals during day sleep and night sleep in preterm and term infants. The maturation of sleep and spontaneous arousals during the day and during the night could reflect the maturation of the circadian clock and, therefore, of the central nervous system in preterm newborns. Studying arousability from day and night sleep could also help in assessing the risk of SIDS at different nychthemeral times in preterm infants.

## 2. Materials and Methods

### 2.1. Study Design and Questionnaires

An Autonomic Baby Evaluation (AuBE) study was conducted to assess the effect of autonomic and sleep maturation on the psychometric development in a cohort of term and preterm newborns at 3 years [28]. The AuBE study was conducted at the Centre Hospitalier Universitaire de Saint-Etienne (France), a level-III university maternity ward including neonatal intensive care units, which oversees 3500 births annually. Inclusions took place over a 24 month period, from September 2009 to September 2011. Pregnancy data (smoking status or nicotine substitute use, alcohol use, drug abuse, beverages containing caffeine consumption) were collected from mothers after birth. Mothers provided at inclusion their age and type of professional activity, which was categorized as without work, manual worker, employee, and executive worker.

### 2.2. Polysomnograms

The same protocol was used as in a previous study on sleep and arousal maturation on day and night in term infants [27].

### 2.3. Psychometric Assessment

A psychometric evaluation using the Wechsler Preschool and Primary Scale of Intelligence (WPPSI-III) [29] was carried out on three-year-olds.

### 2.4. Ethics

The local research ethics committee approved the study (NCT00951860) (CPP: Référence dossier 2009-16, comité de protection des personnes Sud Est I le 11/05/2009) and EUDRACT/2009-A00325-52: autorisation AFSAPS obtenue le 17/07/2009. Written informed consent was obtained at enrolment from the parents. The study was registered on ClinicalTrials.gov (NCT00951860).

### 2.5. Statistical Analyses

The Fisher’s exact, chi-square, and Kruskall–Wallis tests were used for descriptive analyses. An analysis using a mixed model was performed to study the ranks of sleep characteristics accounting for prematurity. Groups were also compared two by two and *p*-values were corrected for multiple comparisons. Statistical significance was defined as a *p* < 0.05. Statistical analyses were performed using SAS software v9.4 (SAS Institute for Data Management, Cary, NC, USA).

In the models built, the response variables were the variables of interest (i.e., the sleep parameters: TST, SE, AS%, QS%, arousals/h, arousals in AS/h, and arousals in QS/h). The explicative variables were time (night/day), maturation (M0/M6), and prematurity (yes/no). Sleep parameters were analyzed one by one using a mixed linear model with each subject considered as a repeated measure, accounting for smoking mothers and child dysmaturity (small for gestational age). The difference in PCA for PSG at M0 and M6 between preterm and term infants were also considered. Interactions between time and maturation and between prematurity and maturation were tested.

## 3. Results

A total of 33 infants (12 preterm and 21 term) were included in the present study. The gestational age, corrected age at M0 and at M6, and birth weight were significantly lower for preterm compared to term children. The proportion of mothers smoking during pregnancy was higher for preterm infants (*p* = 0.04). Premature children were more likely to be small for their gestational age (SGA). The SGA newborns and those from smoking mothers were not the same infants. There was no significant difference between term and preterm infants regarding sex, maternal age during pregnancy, and WPPSI-III characteristics (verbal, performance, and total Intelligence Quotient (IQ) at 3 years; Table 1).

During night sleep, at M0, the preterm newborns slept less compared to the term newborns (TST). At M0 and M6, the preterm infants had a lower SE and a lower total arousal index, more specifically, fewer arousals in AS and more arousals in QS, compared to the term infants. At M6, the preterm infants had a higher QS% and a lower AS% than the term infants (Table 2).

During day sleep, at M0, the sleep stages were similar in the term and preterm newborns. The preterm newborns had a lower arousal index, more specifically, fewer arousals in AS and more arousals in QS, compared to the term newborns. At M6, they had a higher QS%, a lower AS%, and a lower total arousal index compared to the term infants (Table 2).

The adjusted models showed that the QS% was significantly higher at M6 compared to M0 (*p* < 0.0001) and the AS% was significantly lower at M6 compared to M0 (*p* < 0.0001), without a significant effect of time (i.e., day sleep or night sleep). There was no significant difference in QS% between the preterm and term infants (*p* = 0.30), but the significant decrease in AS% was greater in the preterm compared to the term infants (*p* = 0.03). There was a significant interaction between maturation and time (i.e., day sleep and night sleep) from M0 to M6, and TST increased during the night and decreased during the day (*p* < 0.001) without a significant difference between preterm and term infants (Figure 1).

A similar interaction was observed for SE (*p* < 0.001). Significant interactions were also found between maturation and time for the total arousal index (*p* = 0.02) and arousals in AS (*p* = 0.0005), both characterized by a decrease during the day and an increase during the night from M0 to M6. The preterm infants had a lower total arousal index (*p* = 0.01) and arousals in AS (*p* = 0.04), and higher arousals in QS (*p* = 0.02), than the term infants (Figure 2).

## 4. Discussion

With maturation, in both the term and preterm infants, TST and SE decreased during day sleep and increased during night sleep, and the QS% increased while the AS% decreased [30]. As previously reported, we found that the maturation of arousal changed with age differently during night sleep and day sleep [27]. The number of arousals, especially in AS, increased with maturation during night sleep and decreased during day sleep for both the preterm and term infants. This difference in maturation between day sleep and night sleep could be explained by the emergence of a circadian behavioral rhythm during the earliest postnatal period, as reported previously [31,32,33]. This difference between day sleep and night sleep is even more pronounced over time with maturation [18].

The maturation of sleep stages was more pronounced during day sleep (more QS/NREM and less AS/REM) than during night sleep, which is consistent with the hypothesis of the independent development of nervous system states of alertness, as well as REM and NREM sleep in the perinatal period during day and night [22]. Not only was a decrease in the amount of REM sleep observed with age, but also a change in the distribution patterns, i.e., less daytime REM sleep and fewer sleep-onset REM periods [22]. REM sleep at sleep onset is less efficient 6 months after birth than REM within established sleep periods [17]: at this age, an REM episode immediately following sleep onset can be brief and frequently interrupted by other stages or even wakefulness. With maturation, day sleep is less efficient (more fragmented by awakenings) compared to night sleep, which could be related to the decreased number of spontaneous arousals occurring during night sleep, especially in REM sleep. Moreover, diminished arousal responsiveness could be a consequence sleep deprivation. Indeed, in infants who have been sleep-deprived, the number of spontaneous movements decreases while the arousal threshold increases after an auditory challenge in AS [34].

We found that the preterm infants had lower TST and SE at birth during day sleep and night sleep, and a lower SE during night sleep at 6 months. We also found that the preterm infants displayed more QS% and lower AS% in day sleep and in night sleep than term infants at M6, even if they were younger in PGA, as an effect of the accelerated maturation in sleep stages. Hoppenbrouwers et al. did not find any difference in terms of sleep maturation between preterm and term infants [30]. The reason for this discrepancy could be related to the difference in the duration of follow-up (3–4 months vs. 6 months in the present study). The accelerated maturation of preterm infants could be secondary to the stress experienced at birth, which occurs during periods of rapid brain growth, neuronal differentiation, synaptogenesis, and myelination. While all the organs are immature, the brain is one of the most fragile. This fragility is not exacerbated if other stress factors are present in utero, such as maternal smoking, for example [35]. These changes could also be due to the effects of extra-uterine life environmental conditions. Additionally, sleep deprivation or sleep fragmentation in preterm infants could increase sleep pressure and, therefore, the %QS [27,34].

We also found that during their first months of life, the preterm infants had a lower total arousal index during day sleep and night sleep, and fewer arousals in AS during night sleep compared to the term infants, exposing them to a greater risk of SIDS. In fact, several findings indicate that REM sleep constitutes the sleep stage during which the risk of SIDS occurrence is the highest. Indeed, SIDS occurs during the first months of life, when REM sleep dominates. Furthermore, environmental risk factors for SIDS, such as prone sleeping position, prenatal smoking, and high environmental temperature, exert their influence on the deficiency in arousal control during REM sleep [36]. Additionally, the obstruction of the upper airways in future SIDS victims and in healthy infants prenatally exposed to cigarette smoke is also mainly observed during REM sleep [4,37]. REM sleep is characterized by rapid eye movements and low-voltage, fast-wave EEG [38], with both presynaptic and postsynaptic inhibition of afferent neurons, which produces postural hypotonia and contributes to the impairment of ventilatory responses [39]. There is also suspended thermoregulation [40] and a great instability in autonomic functions, exposing infants to bursts of autonomic discharges and rapid cardiorespiratory changes in this sleep stage. This lower arousal index in preterm infants could also be a sign of the delayed maturation of awakening circuits.

However, the preterm infants had a higher arousal index in QS during night sleep at both M0 and M6, and during day sleep at M0, compared to the term infants. We do not have an explanation for those results, but we could relate them to the theory of ‘the resilient brain’ by Parrino et al., according to which the occurrence of awakenings in NREM sleep is an ultimate defense mechanism developed by the organism in order to survive [41]. This theory emerges from the study of awakening reactions, and more specifically from the CAP (cycling alternative pattern) described in NREM. Indeed, a CAP cycle is composed of two phases: phase A (cortical activation EEG), followed by phase B (period of cortical deactivation), lasting at least a minute; the same cycles repeat themselves. These CAPs are interpreted as markers of sleep instability, and are correlated with sleep-related pathologies. In fact, CAPs can reflect a brain reaction to any endogenous or exogenous perturbation during sleep. We do not know whether arousals during QS/NREM sleep in preterm infants could be considered as a phenomenon of resilience designed to regain normal functions (in a few months) after the stress experienced at birth and during early extra-uterine life.

The maturation of arousals was different during day sleep and night sleep. This difference must be kept in mind while interpreting the results of studies on arousals in infants, and could in fact explain the inconsistencies between day and night studies. In fact, studies have shown that infant victims of apparent life-threatening events (ALTE) displayed more spontaneous arousals in day sleep [42] and fewer in night sleep [43] than control infants. In comparison to infants sleeping in the supine position, those sleeping in a prone position display more cortical arousals during day sleep [44], but less during night sleep [45].

Our study has limitations. Only a few preterm and term infants were included in this study. Furthermore, the 24 h PSG was recorded at home at 6 months. In order to study the maturation profile, only the newborns for whom the Electroencephalogram (EEG) was of good quality for both recordings at M0 and M6 were included. Due to the AuBE study schedule, we did not record sleep at 3 months, the age at which the risk of SIDS is maximal. This study should be performed in the future to investigate whether these differences in arousability could explain some of the discrepancies observed between night and day studies. Unfortunately, the presence of obstructive breathing, which is potentially responsible for respiratory-related arousals, could not be assessed. As we know that preterm infants are more at risk of developing obstructive apnea syndrome [46], their lower arousability could expose them to a higher risk of SIDS. Our results should be confirmed by a study including more preterm infants, investigating the respiratory level in a more complete manner, and during the full first year of life (term, M3, M6, and M12). Additionally, some of the preterm infants were born from smoking mothers; this might have impacted on our results. Indeed, exposure to cigarette smoking during prenatal life can alter infant brain structures, potentially inducing some changes in cardiorespiratory and arousal control, via a complex interplay between fetal hypoxia, the absorption of toxins, and metabolic changes. However, in our statistical models, we adjusted for tobacco exposure during pregnancy, and a previous study reported only mild changes between term infants born from smoking and from non-smoking mothers, which were potentially related to mild tobacco exposure [27]. Moreover, since several authors describe the impairment of the autonomic nervous system in preterm infants after birth [47,48], it would be interesting to study the autonomic changes during arousal in term and pretem infants in relation to neurocognitive evolution in further studies. Finally, in this study, we included moderate-to-late preterm (32 to 36 weeks) newborns; this could explain why no difference was found between the preterm and term infants at 3 years through neurocognitive evaluation. This difference in arousability could not be imputed to neurological impairment, as was shown previously, in a longitudinal study including extremely preterm (less than 28 weeks) newborns with impaired neurological development, who had a lower arousability at term compared to those with normal development followed up until 50 months [49]. However, we must note that late-preterm infants were more at risk of SIDS than term infants [26].

## 5. Conclusions

This study showed that preterm infants displayed an accelerated maturation of sleep structure during day sleep and night sleep, with more QS% and less AS% than term infants, which may reflect accelerated brain maturation secondary to stress or environmental exposure after birth. Similarly, there was a difference in arousal maturation during day sleep and night sleep in preterm infants, as shown previously for term infants, which suggests the emergence of a circadian rhythm during the earliest postnatal period. From a practical point of view, we must take into account these differences in maturation between day and night sleep in the interpretation of studies on infants. We also showed that moderate-preterm infants had fewer total arousals and, specifically, more arousals in AS during day sleep and night sleep, exposing them to a higher risk of SIDS. An impairment of the arousal mechanism could also be implicated in the mechanism of SIDS in preterm infants. We must continue to deliver prevention advice on safe sleep for these high-risk groups.

## Figures and Tables

**Figure 1 children-09-00223-f001:**
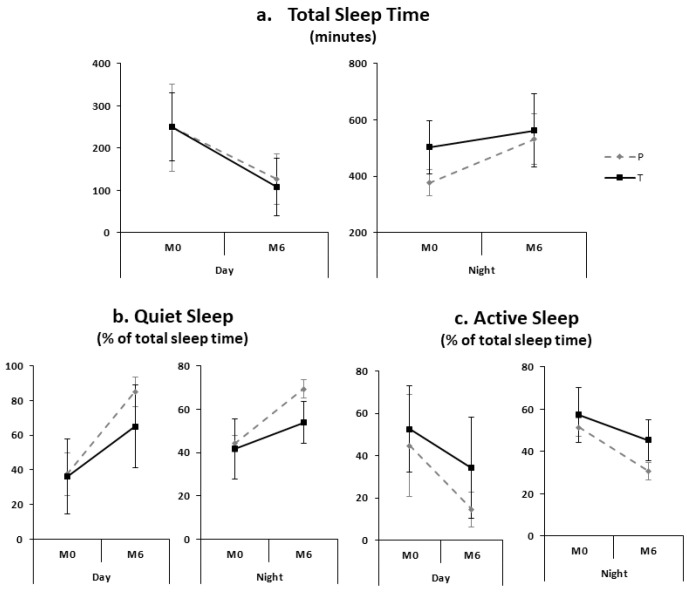
Sleep maturation during day and night sleep in preterm (dotted lines) and term (solid lines) infants.

**Figure 2 children-09-00223-f002:**
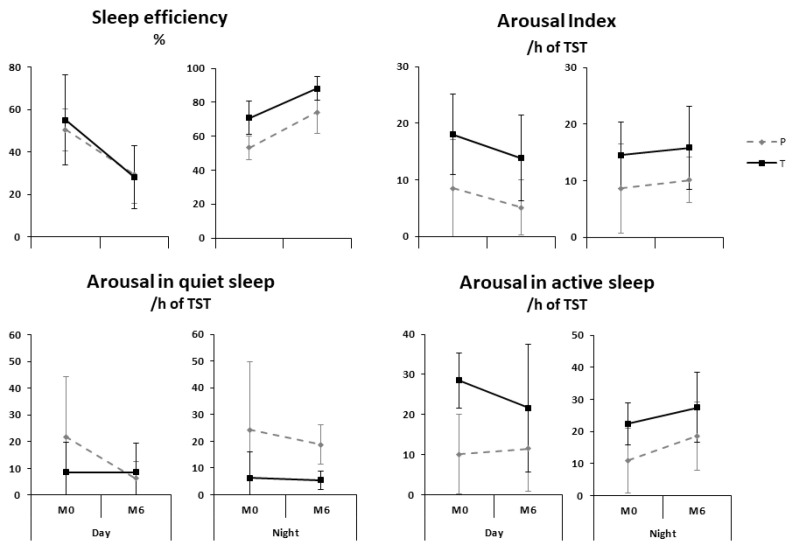
Sleep efficiency and arousal maturation during day and night sleep in preterm (dotted lines) and term (solid lines) infants.

**Table 1 children-09-00223-t001:** Characteristics of the study population.

	Preterm Infants	Term Infants	*p*
*N*	12	21	
Male sex, *n* (%)	8 (67%)	7 (33%)	0.08
Gestational age, weeks, mean ± SD	35.1 ± 2.1	39.8 ± 0.8	<10^−4^
Corrected age at M0, weeks, mean ± SD	37.2 ± 1.6	40.1 ± 1.2	<10^−4^
Corrected age at M6, weeks, mean ± SD	66.4 ± 7.0	70.6 ± 12.0	<10^−4^
Birth weight, g, mean ± SD	2033 ± 598	3319 ± 308	<10^−4^
Immaturity, *n* (%)	3 (25%)	0 (0%)	0.04
Smoking during pregnancy, *n* (%)	3 (25%)	0 (0%)	0.04
Maternal age during pregnancy, years, mean ± SD	32.1 ± 5.3	32.4 ± 4.0	0.81
Verbal IQ at 3 years, mean ± SD	109 ± 17	103 ± 17	0.19
Performance IQ at 3 years, mean ± SD	87 ± 13	86 ± 11	0.60
Total IQ at 3 years, mean ± SD	98 ± 15	93 ± 14	0.20

**Table 2 children-09-00223-t002:** Sleep and arousal characteristics during day and night for term and preterm babies at M0 and M6 recordings.

**NIGHT SLEEP**								
**At M0**					**At M6**			
**mean ± SD**	All	Preterm	Term	*p* value	All	Preterm	Term	*p* value
**TR (min)**	728 ± 94	710 ± 20	738 ± 116	0.07	692 ± 124	720 ± 1.2	676 ± 155	0.82
**TST (min)**	457 ± 101	377 ± 47	503 ± 95.4	0.0005	552 ± 118	532 ± 90	563 ± 131	0.12
**Sleep efficiency (%)**	64.4 ± 12.3	53.2 ± 6.8	70.8 ± 9.9	<10^−4^	82.9 ± 11.5	74.0 ± 12.6	88.1 ± 6.9	0.002
**Quiet sleep (% of TST)**	42.6 ± 11.2	44.1 ± 3.8	41.7 ± 13.8	0.50	59.5 ± 11.1	69.3 ± 4.2	53.9 ± 9.8	<10^−4^
**Active sleep (% of TST)**	55.2 ± 10.8	51.5 ± 4.4	57.3 ± 12.8	0.38	10.0 ± 10.7	30.7 ± 4.2	45.4 ± 9.6	<10^−4^
**Arousal index (/h of TST)**	12.3 ± 7.2	8.6 ± 7.9	14.5 ± 5.9	0.005	13.7 ± 6.7	10.1 ± 4.0	15.8 ± 7.4	0.02
**Arousal in quiet sleep (/h of TST)**	12.8 ± 18.9	24.3 ± 25.3	6.3 ± 9.8	0.0005	10.3 ± 8.3	18.8 ± 7.3	5.5 ± 3.4	<10^−4^
**Arousal in active sleep (/h of TST)**	18.3 ± 9.6	11.0 ± 10.1	22.4 ± 6.6	0.002	24.3 ± 11.5	18.6 ± 10.7	27.5 ± 10.9	0.03
**DAYTIME SLEEP**								
**At M0**					**At M6**			
**mean ± SD**	All	Preterm	Term	*p* value	All	Preterm	Term	*p* value
**TR (min)**	483 ± 140	500 ± 174	474 ± 120	0.39	419 ± 178	444 ± 176	407 ± 182	0.37
**TST (min)**	249 ± 88	249 ± 103	250 ± 81	0.75	115 ± 65	127 ± 60	109 ± 68	0.36
**Sleep efficiency (%)**	53.3 ± 17.9	50.4 ± 10.0	54.9 ± 21.2	0.29	28.5 ± 14.1	29.3 ± 13.5	28.1 ± 14.7	0.93
**Quiet sleep (% of TST)**	36.6 ± 18.6	37.6 ± 12.4	36.0 ± 21.6	0.45	71.6 ± 22.1	85.1 ± 8.5	65.1 ± 23.8	0.02
**Active sleep (% of TST)**	49.8 ± 21.7	44.8 ± 24.0	52.6 ± 20.3	0.19	27.9 ±22.1	14.6 ± 8.3	34.2 ± 23.9	0.02
**Arousal index (/h of TST)**	14.6 ± 8.8	8.5 ± 8.6	18.0 ± 7.1	0.0004	11.0 ± 7.9	5.1 ± 4.9	13.8 ± 7.6	0.004
**Arousal in quiet sleep (/h of TST)**	13.4 ± 17.3	21.8 ± 22.5	8.5 ± 11.4	0.007	7.7 ± 9.5	6.2 ± 6.3	8.5 ± 10.8	0.51
**Arousal in active sleep (/h of TST)**	21.8 ± 12.0	10.1 ± 9.9	28.5 ± 6.9	0.0002	18.3 ± 15.0	11.5 ± 10.6	21.6 ± 16.0	0.12

Data are expressed as mean ± SD. TR: Recording Time, TST: Total Sleep Time.

## Data Availability

Data are not available do to ethical and privacy restrictions: consents forms do not allow data utilization by other research teams. Data may be available on request from the corresponding author and after additional and documented consent of parents involved in this study.

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
