# Peer review of "Maturation of Arousals during Day and Night in Preterm Infants"

_children, 2022, doi:10.3390/children9020223_

Round 1

Reviewer 1 Report

Dear Authors,

In my opinion the theoretical goal is very important and has been largely achieved.

I ran out of practical tips for the announced application goal.
I would expect a few sentences on this: how to apply the results of own research in practice?

Author Response

We thank the reviewer for their comments.

We added in the conclusion:

line 6: We must keep into account these differences in maturation between day and sleep in the intrepretation of studies in infants.

line 10: An impairment in arousal mechanism could also be implicated in the mechanism of SIDS in preterm infants. We must continue to delivery prevention advices on safe sleep in these high-risk group.  

Reviewer 2 Report

Maturation of arousals during day and night in preterm infants by Guyona et al reports on a prospective study involving 12 preterm and 21 term infants who received 24-hour polysomnograms at birth and age 6 months, and a psychometric assessment using the WPPS-III at age 3 years. Their research holds particular interest in SIDS, where it would be difficult to overestimate the importance of sleep studies conducted by Andre Kahn, findings that decades later continue to inform some of the basic hypotheses in SIDS as they pertain to arousal. The research team’s focus on the effects of prematurity on arousal and sleep state organization have further interest in that preterm infants have an increased risk for SIDS.

I found this to be a fine piece of scholarship. The introduction is clear and nicely presented, the methods are sound, results are clearly and fairly stated, and the discussion is careful and well done. I agree with the emphasis on their cortical arousal data as most relevant to their research questions. The size of their study poses limits on power and generalizability in general, though the authors have been open about it. 

The autonomic and cognitive elements to this study are not given much attention. I believe the authors should present their cardiorespiratory data. Moreover, I believe any relationships between cortical arousals and this data should be commented on in the discussion. I think this would be especially important when reviewing the arousal indices and active sleep arousals in the preterm infants.

It is difficult for me to comment on the importance of their findings, but I think this work would be cited for its observations about differential arousal patterns in preterms.

I was distracted by some of the authors’ English language usage and believe the manuscript should be edited from that perspective. I do not believe this undermines their messages, however.

Author Response

We thank the reviewer for his/her comments.

We did not find differences in neurocognitive development between our preterm and term newborns. As explained in the discussion:

"Finally, we included in this study moderate to late preterm (32 to 36 weeks) newborns, this can explain that no difference was found in terms of neurocognitive evaluation between the preterm and term infants at 3 years."

We agree with the reviewer on the interest to study the automomic changes during arousals and we added in the discussion:

Moreover, as several authors mentioned the impairment of autonomic nervous system in preterm infants after birth [51-52], it would be interesting to study the autonomic changes during arousals in term and pretem infants in relation to the neurocognitive evolution in further studies. 

51. Patural, H.; Barthelemy, J.C.; Pichot, V., Mazzocchi, C.; Teyssier, G.; Damon, G.; Roche, F. Birth prematurity determines prolonged autonomic nervous system immaturity. Clin Autonomic Res 2004, 14, 391–395.

52. Horne, R.S.C.; Witcombe, N.B.; Yiallourou, S.; Scaillet, S., Thiriez, G., Franco, P. Cardiovascular control during sleep in infants: implications for the Sudden Infant Death Syndrome. Sleep Med 2010, 11, 615-621.

English correction has been done as requested.

Round 2

Reviewer 2 Report

I was reluctant to make the following points, but the revision created inaccuracies in the introduction and, though the authors adequately addressed my concerns about including observations of arousal changes as they relate to autonomic activity, I was still left left with a question.

  1. Paragraph 1 of the introduction has been revised and the changes introduce inaccuracies. In page 1 line 45, the authors now write that about the association of SIDS with arousal "as suggested by the temporal association between SIDS and sleep periods", but this association has been further developed using infant monitoring data of captured events (see Sridhar or work by Thach) and animal models with altered serotonin, something observed in a high proportion of SIDS brains (Nattie, Dymecki, Cummings, or Kinney). Additionally, on page 2, line 50, the authors describe the Triple Risk Model as "based upon epidemiological studies"—it was not based upon epidemiology but on neuropathological observations that have stood up over time. In line 53 they describe intrinsic vulnerabilities as underlying "prenatal vulnerabilities", which suggests that the defects are developmental in nature though we understand that they may also involve genetic or exposure mechanisms. Then again, in line 54, they state that the vulnerabilities may be caused by premature birth or an adverse intrauterine environment, yet it has been proposed that the preterm birth may itself be reflective of the intrinsic vulnerabilities. These problems were not present in the manuscript's previously reviewed form.
  2. On page 10, the authors have included mention of the importance of autonomic changes as they relate to arousals, which is fine as a future direction. However, their description of their polysomnogram methods in the originally reviewed manuscript suggests that autonomic data was collected (ECG, plethysmography, oxygen saturation). My original question was why this data was not included. As a reviewer, I had wanted this clarified.

Author Response

We thank the reviewer for his/her comments.

Dear Reviewer,

The editor asked us to modify and shortened the introduction for editing process. We also cut the paragraph with polysomnography recording and analyses. As you wished, we came back and put the first version of the introduction.

Indeed, we recorded during PSG: 1 electrocardiogram, chest and abdominal respiratory movements by inductance plethysmography, as well as noninvasive arterial oxygen saturation using an oximetry probe placed on the foot. The quality of respiratory and saturation parameters was non-optimum. We preferred to analyse ECG at different times to evaluate the maturation of ANS in infants (Patural H et al. Helyion 2019). We planned to submit this year an article on the maturation of ANS according to the sleep stages in relation to cognitive development in term and preterm infants.